# Nonlocality activation in a photonic quantum network

Luis Villegas-Aguilar [1], Emanuele Polino[1], Farzad Ghafari [1], Marco Túlio Quintino [2], Kiarn T. Laverick[3], Ian R. Berkman [4], Sven Rogge [4], Lynden K. Shalm[5], Nora Tischler [1] ✉, Eric G. Cavalcanti [3] ✉, Sergei Slussarenko [1] & Geoff J. Pryde [1]

Bell nonlocality refers to correlations between two distant, entangled particles that challenge classical notions of local causality. Beyond its foundational significance, nonlocality is crucial for device-independent technologies like quantum key distribution and randomness generation. Nonlocality quickly deteriorates in the presence of noise, and restoring nonlocal correlations requires additional resources. These often come in the form of many instances of the input state and joint measurements, incurring a significant resource overhead. Here, we experimentally demonstrate that single copies of Bell-local states, incapable of violating any standard Bell inequality, can give rise to nonlocality after being embedded into a quantum network of multiple parties. We subject the initial entangled state to a quantum channel that broadcasts part of the state to two independent receivers and certify the nonlocality in the resulting network by violating a tailored Bell-like inequality. We obtain these results without making any assumptions about the prepared states, the quantum channel, or the validity of quantum theory. Our findings have fundamental implications for nonlocality and enable the practical use of nonlocal correlations in real-world applications, even in scenarios dominated by noise.

Quantum entanglement and Bell nonlocality[1], though intimately related, are fundamentally inequivalent manifestations of quantum theory. All pure entangled states display nonlocal correlations[2], but quantum systems are invariably subject to noise in the real world. The presence of noise degrades the quality of nonclassical correlations, as evidenced by the existence of entangled Bell-local states—mixed entangled states[3,4] that cannot display any nonlocality in the standard Bell scenario. The motivation behind the study of nonlocality is not limited to foundational insights into quantum theory since nonlocal correlations are at the heart of many quantum technologies[5].

A significant discovery in counteracting the effects of noise was that nonlocality can be activated: entangled states that cannot display nonlocal correlations in any standard Bell test can recover their nonlocality when using additional resources[6]. For some restricted families of states, a single copy of a Bell-local state can be activated using more intricate measurement procedures[7–11]. In cases when more than one copy of the state is available, proposed activation protocols require performing joint measurements on several quantum states distributed between two[12–14] or multiple[15,16] spatially separated parties. Scenarios involving many parties provide considerably stronger generalizations of Bell nonlocality[17] with the potential to yield powerful activation

[1]Centre for Quantum Dynamics and Centre for Quantum Computation and Communication Technology, Griffith University, Yuggera Country, Brisbane, QLD 4111, Australia. [2]Sorbonne Université, CNRS, LIP6, Paris F-75005, France. [3]Centre for Quantum Dynamics, Griffith University, Yugambeh Country, Gold Coast, QLD 4222, Australia. [4]Centre for Quantum Computation and Communication Technology, School of Physics, The University of New South Wales, Sydney, NSW 2052, Australia. [5]National Institute of Standards and Technology, 325 Broadway, Boulder, CO 80305, USA. ✉e-mail: n.tischler@griffith.edu.au; e.cavalcanti@griffith.edu.au

schemes[18,19]. Multi-copy approaches for activation, however, are presently unfeasible as the number of necessary copies of the states increases rapidly with noise[20]. Despite the importance of nonlocality in quantum foundations and technologies, a robust and resource-efficient activation is yet to be realized.

Here we demonstrate an experimental activation of nonlocality in a photonic quantum network using a single copy of the target state per experimental round. We achieve this by departing from typical correlation scenarios in networks[21]–where independent parties are connected by independent sources of entanglement–towards scenarios with a more general causal structure[22,23], enabling distinct forms of quantum advantages in networks. We employ a quantum channel[24] that broadcasts part of an entangled Bell-local state to two spatially separated parties, embedding a bipartite quantum state into a three-party network, as shown in Fig. 1. Importantly, our activation is certified through a rigorous and robust statistical analysis of the Bell locality of the original bipartite states. We present a computationally efficient method to prove the existence of local hidden variable (LHV) models for general quantum states. In this manner, we prepare certified Bell-local states, which, after the activating procedure, unambiguously show the emergence of nonlocality from the observed network statistics. Our results are obtained exclusively from experimental data, without making any assumptions about the prepared states or quantum channel.

From a fundamental point of view, we demonstrate that the nonlocal behavior of Bell-local states can be unveiled when they are integrated into larger networks. This illustrates a form of non-classicality within networks that extends beyond the conventional notions of network Bell nonlocality[17]. On a practical note, our results open up possibilities for quantum applications involving noisy states. This recovers the potential of nonlocality-based applications in more realistic contexts, encompassing tasks such as secure communications[25], generating randomness[26], or certifying entanglement within a network[27,28].

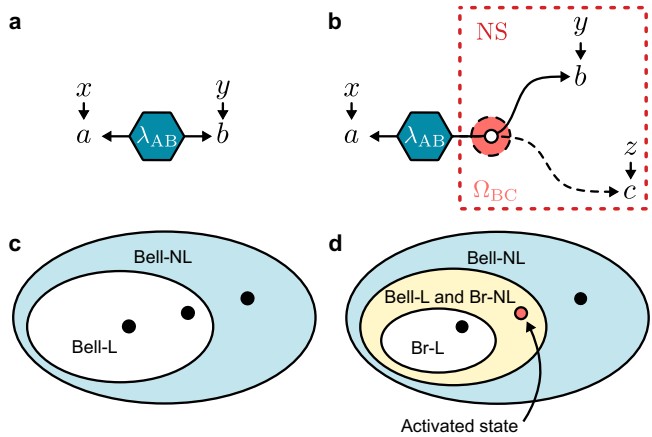

**Fig. 1 | Nonlocality scenarios. a** Causal structure for the standard Bell scenario, in which a classical resource $\lambda_{AB}$ is shared between two parties. $\lambda_{AB}$ is responsible for the observed joint correlations of measurement outcomes $a$ and $b$ given inputs $x$ and $y$, respectively. **b** Broadcast scenario with three parties, where a classical bipartite resource $\lambda_{AB}$ is shared between one party and a broadcast channel $\Omega_{BC}$. Measurement nodes receive inputs $x$, $y$ and $z$, respectively, yielding outcomes $a$, $b$ and $c$. The broadcast parties are subject only to no-signaling (NS) constraints. **c, d** Schematic representation of the membership of quantum states within different correlation sets for the bipartite Bell (**c**), and tripartite broadcast (**d**) scenarios. Bell-NL (Br-NL) signify Bell-nonlocal (broadcast-nonlocal) correlations; states in the Bell-L (Br-L) set are Bell-local (broadcast-local), admitting a local hidden variable in their respective scenario. The intermediate yellow region in (**d**) represents the set of Bell-local states that can be activated in the broadcast scenario.

## Results

### From Bell to broadcast nonlocality

The differences between testing nonlocality in a typical Bell scenario and our three-node quantum network are highlighted in Fig. 1. In the simplest Bell scenario, a bipartite source $S_{AB}$ distributes a pair of systems among two distant parties, Alice and Bob. These parties perform measurements $x$ and $y$ on their local subsystem, obtaining binary outcomes $a$ and $b$, respectively. If the correlations arising from the measurement outcomes are compatible with the causal structure of Fig. 1a–under the assumption that $S_{AB}$ is a source of classical shared randomness $\lambda_{AB}$–then they can be described with an LHV model of the form

$$p(a,b|x,y) = \int d\lambda_{AB}\,p(\lambda_{AB})\,p_A(a|x,\lambda_{AB})\,p_B(b|y,\lambda_{AB}), \quad (1)$$

for some distribution $p(\lambda_{AB})$. If the correlations cannot be described this way, they are said to be Bell-nonlocal. This is witnessed by the violation of suitable Bell inequalities.

The three-node network depicted in Fig. 1b also features a single source of two particles. This scenario, however, incorporates an additional channel that applies the transformation $\Omega_{BC}$ on part of the initial state. The effect of this transformation is to distribute the information encoded in one of the particles to two additional parties, Bob and Charlie. One can introduce an additional LHV associated with the channel, but the resulting statistics would be equivalent to a standard tripartite Bell scenario (see Methods and Ref. 24). Conversely, when no constraints are placed on the channel other than the preparation of no-signaling resources[29], an LHV model for the source $S_{AB}$ in this network can be written as

$$p(a,b,c|x,y,z) = \int d\lambda_{AB}\,p(\lambda_{AB})\,p_A(a|x,\lambda_{AB})\,p_{BC}^{NS}(b,c|y,z,\lambda_{AB}). \quad (2)$$

Here, $p_{BC}^{NS}(b,c|y,z,\lambda_{AB})$ indicates that the only constraint for the correlations shared between Bob and Charlie is that they must be no-signaling, conditioned on the source preparing the classical state $\lambda_{AB}$. This assumption follows the theory-independent spirit of Bell's theorem, as it does not rely on the validity of quantum mechanics, and has the critical consequence that any nonlocality observed from the correlations arising in the three-party scenario must have originated from the initial source $S_{AB}$.

The certification of nonlocality in this setting comes as a tailored causal compatibility inequality[24] for the distribution $p(a, b, c|x, y, z)$ in the form

$$\begin{aligned}
\mathcal{I}_B = &\langle A_0 B_0 C_0 \rangle + \langle A_0 B_1 C_1 \rangle + \langle A_1 B_1 C_1 \rangle - \langle A_1 B_0 C_0 \rangle \\
&+ \langle A_0 B_0 C_1 \rangle + \langle A_0 B_1 C_0 \rangle + \langle A_1 B_0 C_1 \rangle - \langle A_1 B_1 C_0 \rangle \\
&- 2\langle A_2 B_0 \rangle + 2\langle A_2 B_1 \rangle - 4 \leq 0,
\end{aligned} \quad (3)$$

with $\langle A_x B_y C_z \rangle = \Sigma_{a,b,c=0,1}(-1)^{a+b+c}\,p(a,b,c|x,y,z)$ and analogously for the two-party correlator terms. A violation of this inequality implies the failure of equation (2), without any assumption about the type of resources produced by the broadcasting device–whether classical, quantum, or described by some general probabilistic theory[30]. In this sense, the violation of inequality (3) can be understood as ruling out that the source $S_{AB}$ is classical, even while allowing any generalized causal model with the causal structure of Fig. 1b[23].

### Nonlocality activation

For the task of activating nonlocality, we are interested in whether: (i) we can observe tripartite quantum correlations that do not admit a description as in equation (2); and (ii) the bipartite state prior to broadcasting is local in the standard Bell scenario of Fig. 1a. That is, we need to certify that all correlations supported by this state admit an LHV model of the form (1). A simultaneous validation of both would be conclusive proof for the activation of nonlocality.

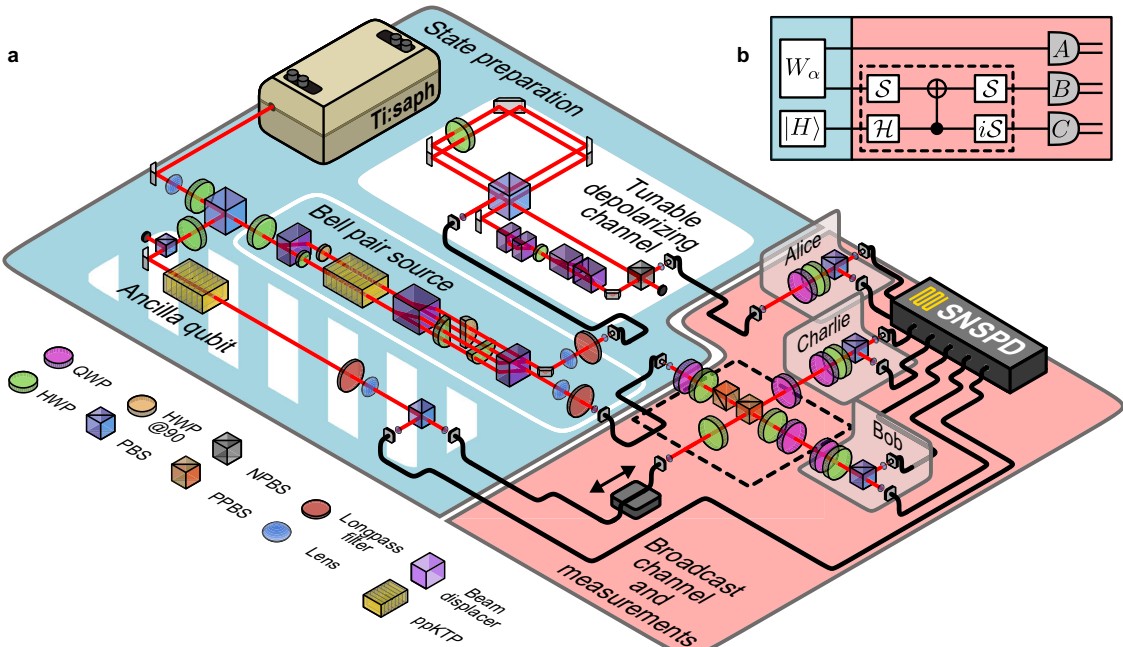

**Fig. 2 | Schematic overview of the experiment. a** The experimental setup comprises state preparation (blue area) and a broadcast channel with a photon measurement stage (red area). We constructed two single-photon pair sources by pumping two identical periodically poled potassium titanyl phosphate crystals (ppKTP) with a modelocked laser centered at 775 nm. Each source produced frequency-degenerate single photons at 1550 nm via type-II spontaneous parametric downconversion (SPDC). One source (solid white rim) generated a maximally entangled state, which was controllably depolarized (solid white background) to tune the parameter $\alpha$ and prepare the state $W_\alpha$ in equation (4). A second source (striped white background) produced a heralded ancilla photon, initialized to $|H\rangle$. After the broadcast channel (dashed black box), the resulting state was transmitted to three spatially separated parties for projective polarization measurements. Single photons were detected with superconducting nanowire single-photon detectors (SNSPD), and a time-to-digital converter identified four-fold coincidences within a 1 ns window. **b** Quantum circuit for activation. The broadcast channel, highlighted by the dashed box, consisted of a C-NOT, Hadamard ($\mathcal{H}$), and $\mathcal{S} = \left(\begin{smallmatrix} 1 & 0 \\ 0 & 1 \end{smallmatrix}\right)$ gates, where $i$ represents an additional $\pi/2$ phase. $A$, $B$, and $C$ indicate local projective measurements performed by parties Alice, Bob, and Charlie. QWP, quarter-wave plate; HWP, half-wave plate; PBS, polarizing beamsplitter; NPBS, non-polarizing beamsplitter; PPBS, partially polarizing beamsplitter.

A positive answer to the first question is obtained whenever a violation of inequality (3) is observed. Answering the second question —a rigorous demonstration that an arbitrary state belongs to the class of Bell-local states—is also a difficult task[31]. While the locality bounds for some classes of quantum states have been extensively studied, it is unclear to what degree these findings can be extended to experimentally prepared systems. Experimental states inevitably deviate from theoretical targets, yet a typical approach is to make assumptions about the type of state at hand and draw conclusions based on common benchmarks like quantum state fidelity, which can be problematic[32]. To tackle this problem, we provide a computational method for constructing LHV models for generic quantum states under general dichotomic measurements, i.e., general two-outcome Positive Operator-Valued Measures (POVMs). Conceptually, our algorithm can be understood as deriving new LHV models for generic quantum states that are close to some reference local state. This involves two steps. We first perform state tomography to obtain a density matrix $\rho_{\exp}$ that best describes our experimental states. Then, we verify the presence of an LHV model for $\rho_{\exp}$ by leveraging existing LHV models of particular quantum states[33] via an efficient certification protocol (see Methods). The algorithm is not restricted to specific families of states; rather, it is designed to be applicable to general quantum states.

Our experimental demonstration of activation employed a photonic setup, as shown in Fig. 2. To successfully implement our three-photon activation protocol, we had to meet strict technological prerequisites, including the use of high-fidelity heralded single-photon and entangled photon-pair sources and a high-quality broadcast channel. These are discussed in depth in the Methods. We used two independent photon-pair sources to generate the required single photons, encoding information in the polarization degree of freedom, such that $|0\rangle \equiv |H\rangle$ and $|1\rangle \equiv |V\rangle$. One photon source was designed to generate the two-qubit isotropic state

$$W_\alpha = \alpha|\Phi^+\rangle\langle\Phi^+| + (1-\alpha)\mathbb{I}_4/4, \qquad (4)$$

where $|\Phi^+\rangle = (|HH\rangle + |VV\rangle)/\sqrt{2}$ is a maximally entangled state and $\mathbb{I}_4/4$ is the maximally mixed state. Here, the parameter $\alpha \in [0,1]$ is the pure-state fraction of the state, which cannot display Bell nonlocality for $\alpha < 0.6875$ under dichotomic measurements (see Ref. 33 and Methods). For general measurements, the current known bound is $\alpha < 0.5$[34,35].

We prepared six experimental states $\rho_{\exp}$ and their measured fidelity, defined as $\mathcal{F} = \mathrm{Tr}\left(\sqrt{\sqrt{\rho_{\exp}} W_\alpha \sqrt{\rho_{\exp}}}\right)^2$, with the nearest $W_\alpha$ state were all $\mathcal{F} > 0.991$ (see Supplementary Table 1).

These values are on par with the highest reported fidelities for two-qubit isotropic states to date[36]. This state was initially shared between Alice and Bob, and the design of the source allowed us to precisely tune the amount of mixture in the state using a controllable depolarizing channel on Alice's qubit (see Methods for details). An additional source was used to generate a heralded single photon as an ancillary resource for the broadcast channel.

The channel for activation included a nondeterministic controlled-NOT (C-NOT) gate[37], which relied on nonclassical Hong-Ou-Mandel (HOM)[38] interference between photons from different sources. The prerequisite for activation in the broadcast scenario is that the broadcast parties satisfy no-signaling constraints. This condition was experimentally enforced by encoding qubits in different photons sent to spatially separated parties. Each party performed local projective

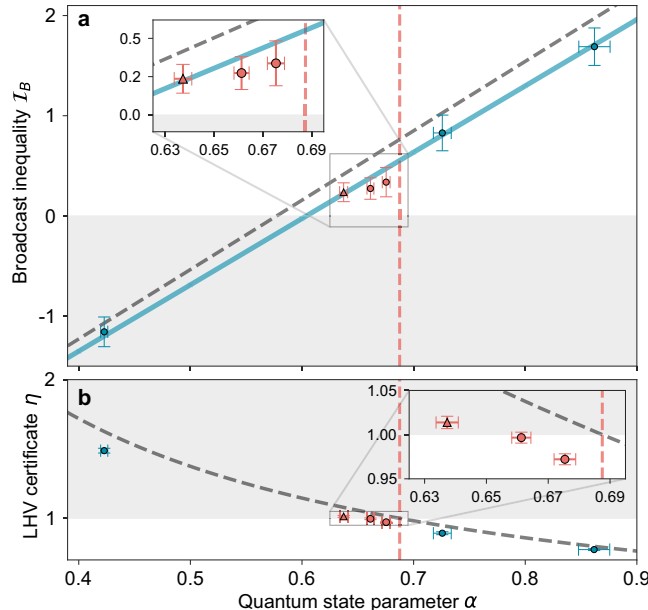

**Fig. 3 | Experimental activation of nonlocality as a function of the quantum state parameter $\alpha$. a** Results for the broadcast inequality $\mathcal{I}_B$ of equation (3). Diagonal lines represent theoretical predictions for ideal (dashed gray) and experimentally observed (solid blue) two-photon interference. Red and blue data points indicate activatable and non-activatable states, respectively. A red triangle symbolizes certified nonlocality activation. The vertical dashed line shows the current Bell-local upper bound ($\alpha \le 0.6875$) for isotropic states under projective measurements in the two-party scenario. The gray area indicates the classical region above which broadcast nonlocality is observed. **b** Locality test for the experimental bipartite states. The dashed curve depicts the certificate results for an ideal isotropic state. Values of the certificate $\eta \ge 1$, shown in the gray area, guarantee the existence of an LHV model for the corresponding quantum state for two-outcome POVMs. Error bars are ±1 standard deviations in total. Uncertainties in $\mathcal{I}_B$ arise from Poissonian statistics, whereas the uncertainties in $\eta$ and $\alpha$ are calculated from Monte Carlo simulations of the different $\rho_{exp}$ that include Poissonian photon-counting noise and systematic errors in measurements (see Methods).

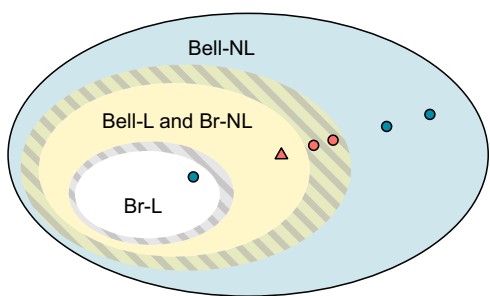

**Fig. 4 | Illustration of the experimental states within the hierarchy of correlations.** The activation of nonlocality is certified for any experimental state that is rigorously proven to belong to the set producing both Bell-local and broadcast-nonlocal correlations, as illustrated by the triangle symbol. Striped areas represent uncertainties for determining the boundaries between correlation sets.

measurements on their respective photon, and data was recorded as fourfold coincidences.

In Fig. 3a, we present our experimental test of the inequality (3), along with theoretical predictions, for a set of states with varying degrees of noise.

The observed experimental values are well captured by the predictions (Fig. 3a, solid diagonal line) derived from a theoretical model

that considers non-ideal HOM interference (which introduces unwanted mixed terms to the final target state) and errors in the performed measurements. We also include predictions for the case of ideal interference (Fig. 3a, dashed diagonal line).

For all the experimental states $\rho_{exp}$, except for the case of the lowest $\alpha$ value, we measured a value of $\mathcal{I}_B > 0$ by at least two standard deviations, representing a clear violation of the classical limit. In particular, three experimental states (Fig. 3a, inset) have an associated value of $\alpha \le 0.6875$, the current known upper bound for projective LHV models of the isotropic state $W_\alpha$[33] (Fig. 3, vertical dashed line). States with larger $\alpha$ also violate the broadcast inequality, but since they can additionally violate a standard two-party Bell inequality under projective measurements, they are not activated.

A definitive demonstration for activation must refrain from making the unrealistic assumption that the experimental states precisely match the form of ideal states. In this spirit, we assess the Bell locality of our original bipartite states via the previously introduced algorithm. We plot these results in Fig. 3b versus the ideal state parameter $\alpha$. A value for the certificate parameter $\eta = 1$ ascertains the existence of an LHV model for the respective state. Values beyond this ($\eta > 1$) indicate that the LHV model is robust against white noise. Here too, we present the certificate results for the case of the ideal isotropic state as a dashed curve, recovering the locality of the state up to the current $\alpha \le 0.6875$ bound. Of the three broadcast states shown in the inset of Fig. 3a, one, depicted by a red triangle, is certifiably activated. The inset of Fig. 3b emphasizes this further. In this case, two of the three previously mentioned states (red circles) yield outcomes that fall below the certificate threshold. These results underscore the necessity of performing such a rigorous locality analysis: even if the associated values of $\alpha$ suggest that the states are Bell-local, one cannot assume this to be the case. At this point, it is important to stress that one should avoid interpreting a value of $\eta < 1$ as an indication of nonlocality in the causal scenario of Fig. 1a; instead, it simply conveys that the Bell locality of the state cannot be conclusively verified. We further tested the two remaining states (red circles) numerically against standard bipartite Bell inequalities[39], which failed to violate the local bounds (see Supplementary Note 1). In this way, one is able to exclude the possibility of them being trivially Bell-nonlocal. We summarize the experimental outcomes in Fig. 4, symbolizing the membership of our experimental states to different correlation sets.

## Discussion

Within the rapidly developing landscape of quantum information, nonlocal correlations lay the foundation for new theoretical and technological discoveries. The original scenario that Bell envisioned was a catalyst for decades of intense research on nonlocality. Now, with the advent of quantum networks, we can explore these correlations in a broader and richer context. Here, we have experimentally demonstrated that nonlocality, as a resource, can be accessed beyond the standard noise limits that are present for standard scenarios involving two parties. To achieve a fully loophole-free implementation, the key assumption to eliminate is the fair-sampling assumption. This would require increasing the overall efficiency, currently limited by the probabilistic implementation of the channel.

We note that, although stronger examples of nonlocality activation are known in multi-copy settings[13,15], this can be prohibitively hard to achieve in practice when dealing with large ensembles of distributed, independent copies of a quantum state. For instance, to achieve an activation under similar noise conditions (i.e., for $\alpha \sim 0.64$), one would require at least $N = 21$ copies of the isotropic state in a star network configuration[18]. For up to $\alpha \sim 0.6875$, $N \ge 10$ copies are still needed[40].

Unlike earlier works on multipartite nonlocality that mainly focused on scenarios consisting only of sources and measurements[5,17], our experimental demonstration reveals the potential for unlocking

further advantages in networks by incorporating an intermediate quantum hub: a node with quantum inputs and many quantum outputs. These results represent a demonstration of nonlocality in more general quantum networks, where the sources are taken to be classical, but the only limitations on any intermediate channel are general no-signaling resources. If these are constrained to allow quantum correlations, the noise tolerance in such network scenarios could be increased even further, while still allowing for fully device-independent (but no longer theory-independent) protocols. One example is entanglement certification[28], where the inclusion of broadcast channels was predicted to significantly improve over standard methods. Incorporating hybrid assumptions into network scenarios[24,41,42] enables key insights into nonlocality tasks. Characterizing correlations in these hybrid scenarios will become increasingly essential as future quantum networks naturally expand in size and complexity.

## Methods

### Classical model in the broadcast scenario

For the scenario in Fig. 1b, one takes the source $S_{AB}$ to be an LHV, as denoted by $\lambda_{AB}$. In a quantum mechanical description, the final joint probability distribution is given by the Born rule

$$p(abc|xyz) = \text{Tr}\left(A_{a|x} \otimes B_{b|y} \otimes C_{c|z}\, \rho_{ABC}\right), \qquad (5)$$

where $\rho_{ABC}$ represents the resulting state after the application of the broadcast channel $\Omega_{BC}$ on half of the input state. If the broadcast channel were assumed to produce an additional classical resource described by a hidden-variable $\lambda'$, which is, in turn, dependent on $\lambda_{AB}$, the distribution could be decomposed as

$$p(a,b,c|x,y,z) = \int d\lambda_{AB}\, p(\lambda_{AB})\, p_A(a|x,\lambda_{AB})\, \tilde{p}_B(b|y,\lambda_{AB})\, \tilde{p}_C(c|z,\lambda_{AB}), \qquad (6)$$

where $\tilde{p}_B(b|y,\lambda_{AB}) = \int d\lambda'\, p_B(b|y,\lambda')\, p_B(\lambda'|\lambda_{AB})$ (and similarly for $\tilde{p}_C$).

The model in equation (6) is equivalent to a standard tripartite Bell-local model, and its violation can be obtained even with a classical $\lambda_{AB}$ (e.g. if the channel prepares a maximally entangled state). Thus, the violation of a standard tripartite Bell inequality cannot be used to rule out a classical description of $S_{AB}$. By relaxing the constraint on $\Omega_{BC}$, allowing it to prepare general no-signaling resources[24], one obtains the decomposition shown in equation (2). Any violation of this then ensures that $S_{AB}$ cannot be described as an LHV.

The no-signaling condition between the broadcast parties is formalized by

$$\sum_b p_{BC}^{NS}(b,c|y,z,\lambda_{AB}) = \sum_b p_{BC}^{NS}(b,c|y',z,\lambda_{AB}) \\ \forall y,y',z,\lambda_{AB}, \qquad (7)$$

$$\sum_c p_{BC}^{NS}(b,c|y,z,\lambda_{AB}) = \sum_c p_{BC}^{NS}(b,c|y,z',\lambda_{AB}) \\ \forall y,z,z',\lambda_{AB}. \qquad (8)$$

This is a weak condition applicable to the channel and is motivated by the assumption that $B$ and $C$ are causally disconnected parties, as per the causal diagram in Fig. 1b. We reiterate that this is not a classicality assumption on the channel—on the contrary, it is even allowed to produce post-quantum resources like PR-boxes[29]. The classicality of $\lambda_{AB}$ is a condition imposed on the source, not on the channel.

### Certifying that states have an LHV model

Our algorithm builds on the conceptual framework introduced in Refs. 43,44 and reviewed in Ref. 45. It incorporates the existence of local models for specific entangled states to certify an LHV model for

general quantum states and general two-outcome POVMs, which are a superset of projective qubit measurements. First, we consider a $d \times d$-dimensional bipartite quantum state $\rho$ with an LHV model for all $n$-outcome POVMs. Let $\Lambda$ be a positive trace-preserving, linear map that acts on a $d$-dimensional system. A positive map is defined such that, for all positive semidefinite states $\sigma$, $\Lambda(\sigma) \geq 0$. If $[\mathbb{I}_d \otimes \Lambda](\rho)$ is a valid quantum state, then it will also have an LHV model for all $n$-outcome POVMs (a detailed proof is provided in Supplementary Note 2). For LHV extension methods, it is sufficient that $\Lambda$ is a positive (but not necessarily completely positive) map[46]. Following this, a suitable choice of $\rho_{LHV}$ is needed. To this end, we use the recent results of Ref. 33 that prove the existence of an LHV model for the two-qubit isotropic state $W_{0.6875}$ under projective measurements. Furthermore, since extremal two-outcome qubit measurements are projective[47], this state also has an LHV model for all two-outcome POVMs. The selection of $\rho_{LHV} = W_{0.6875}$ is thus naturally motivated because our experimentally prepared states are, by design, very close to this family of states.

An additional step is required to find an LHV model for the state $\rho_{exp}$ obtained via quantum state tomography. Explicitly, we use the convexity of the set of states admitting an LHV model for $n$-outcome measurements $\mathcal{L}$. That is, if $\rho_1, \rho_2 \in \mathcal{L}$ then $\rho = q\rho_1 + (1-q)\rho_2 \in \mathcal{L}$, where $q \in [0,1]$. This fact allows us to certify a larger space of states, expanding our search further by searching for a map $\Lambda$ and a local state $\rho_2$ such that:

$$\rho_{exp} = q[\mathbb{I}_2 \otimes \Lambda](W_{0.6875}) + (1-q)\rho_2. \qquad (9)$$

The last step is to select a set of LHV states to explore in the context of $\rho_2$. We opted for the set of separable states, which are readily and fully characterized for bipartite qubit-qubit states using the Positive Partial Transpose (PPT) criterion[48]. According to the PPT criterion, a qubit-qubit state is deemed separable if and only if its density matrix $\rho$ satisfies the condition $\rho^{T_2} \geq 0$, where $T_i$ denotes the partial transpose operation on the $i$th system. The considerations discussed above lead us to the formulation of the following optimization problem:

$$\max \eta \qquad (10)$$

such that:

$$\eta \rho_{exp} + (1-\eta)\frac{\mathbb{I}_4}{4} = q[\mathbb{I}_2 \otimes \Lambda](W_{0.6875}) + (1-q)\rho_{ppt} \qquad (11)$$

$$0 \leq q \leq 1 \qquad (12)$$

$$\rho_{ppt} \geq 0, \ \text{Tr}(\rho_{ppt}) = 1, \ \rho_{ppt}^{T_2} \geq 0 \qquad (13)$$

$$\rho \geq 0 \ \Rightarrow \ \Lambda(\rho) \geq 0 \ (\Lambda \text{ is a positive map}) \qquad (14)$$

$$\text{Tr}[A] = \text{Tr}[\Lambda(A)] \ \forall A \ (\Lambda \text{ is trace preserving}). \qquad (15)$$

A subtle fact to note is that this problem is not strictly equivalent to the one we described earlier in equation (9). Instead, we ask a slightly modified question: What is the minimum amount of white noise that needs to be added to $\rho_{exp}$ until we have a state admitting an LHV model for two-outcome measurements? When the optimization returns a value of $\eta \geq 1$, we can certify that our experimental state has an LHV model for all dichotomic measurements; otherwise, the results are inconclusive.

Our algorithm can be, additionally, computed efficiently. Since the positive map $\Lambda$ is a qubit-qubit map, it can be decomposed as $\Lambda = \Lambda_{CP}^1 + \Lambda_{CP}^2 \circ T$, where $\Lambda_{CP}^1$ and $\Lambda_{CP}^2$ are completely positive and $T$ is the transposition map[48]. It is then possible to use the Choi-

Jamiołkowski isomorphism[49] to phrase this optimization routine as a semidefinite program (SDP), which belongs to a class of optimization problems that can be efficiently solved using precise and efficient methods. Our algorithm takes advantage of existing LHV models for specific states, resulting in a substantial improvement in computational efficiency. Given that an LHV model for $W_{0.6875}$ is guaranteed to exist[33], we exploit this result to efficiently derive new LHV models for general states close to $W_{0.6875}$. While the method in Ref. 33 required around a month to execute on a powerful 64-core computer, our approach can find an LHV model for $\rho_{exp}$ in less than one second using a standard personal computer.

### Broadcast nonlocality activation and POVMs

Our algorithm establishes that the experimentally obtained state $\rho_{exp}$ admits an LHV for all projective measurements and, consequently, for all two-outcome POVMs. A general proof for arbitrary POVMs is still an open question, as this is a much more challenging task even for very well-studied states[34,35].

This becomes particularly relevant as one may consider the case of Bob and Charlie as grouped together, acting collectively. Under this assumption, the broadcast channel $\Omega_{BC}$ and local projective measurements performed by $B$ and $C$ may be reinterpreted as an effective POVM with more than two outcomes. It could then be argued that the activation observed in our experiment can be exclusively attributed to this more general measurement being performed on part of the state and not due to the causal structure considered. To address this, we further analyzed this scenario within the $A$ and $BC$ bipartition.

Formally, we considered the behavior

$$p(a,b,c|x,y,z) = \mathrm{Tr}\left(\rho_{ABC} A_{a|x} \otimes M_{bc|yz}\right)$$
$$= \mathrm{Tr}\left(\rho_{AB} A_{a|x} \otimes \Omega^\dagger_{BC}(M_{bc|yz})\right), \quad (16)$$

where $\Omega^\dagger$ is the adjoint map of $\Omega$ and $\Omega^\dagger_{BC}(M_{bc|yz})$ denotes a valid qubit 4-outcome POVM.

Using standard semidefinite programming methods for quantum steering and joint measurability[50], we show that the effective POVMs performed in our experiment (corresponding to the measurements in Table 1 along with the adjoint of the isometry defined in equation (18)) have a white noise robustness of 0.7746. In other words, when such measurements are performed by parties $B$ and $C$ on part of the two-qubit state $W_\alpha$ for $\alpha \le 0.7746$, the resulting assemblage is unsteerable. Finally, we reconstructed the experimental assemblage

$$\sigma_{bc|yz} := \mathrm{Tr}_B(\rho_{exp} \Omega^\dagger(M_{bc|yz})) \quad (17)$$

and numerically verified that this assemblage is also unsteerable and thus can only lead to Bell-local behaviors, regardless of what measurements Alice performs (see Code availability statement for the full code). This result establishes unambiguously that the observed nonlocality in this work is not a consequence of transitioning from projective measurements to POVMs, but due to the broadcasting causal structure. Indeed, it has been shown that broadcasting scenarios can allow for activation even for states that are known to be Bell-local with respect to arbitrary POVMs[28].

### Table 1 | Optimal projective measurement settings for non-locality activation

| Alice | Bob | Charlie |
|---|---|---|
| $A_0 = \frac{-\sigma_X - \sigma_Z}{\sqrt{2}}$ | $B_0 = \frac{\sqrt{2}\sigma_X + \sigma_Y}{\sqrt{3}}$ | $C_0 = \sigma_Z$ |
| $A_1 = \frac{\sigma_X - \sigma_Z}{\sqrt{2}}$ | $B_1 = \frac{\sqrt{2}\sigma_X - \sigma_Y}{\sqrt{3}}$ | $C_1 = \sigma_X$ |
| $A_2 = -\sigma_Y$ | | |

The question of whether non-projective POVMs are useful for revealing the nonlocality of states that are otherwise local under projective measurements remains a crucial open problem in quantum information science. In the case of some specially tailored Bell inequalities, POVMs may be used to violate it more with more states[51]. But, in the context of EPR steering, equivalence between POVMs and projective measurements has been established for two-qubit Werner states[34,35]. There is also evidence suggesting that the most incompatible sets of qubit POVMs are always projective[52].

### Photon sources

The single photons used in the protocol were generated via type-II SPDC. A mode-locked Ti:sapphire laser at 775 nm was used to produce 1 ps pulses with a repetition rate of 80 MHz at 200 mW of power. The pump pulses were split into two beams with a half-wave plate (HWP) and polarizing beam splitter (PBS), before pumping two separate photon sources with identical ppKTP crystals. The first source, based on the design of Refs. 36,53, embedded one of the ppKTP crystals inside of a beam displacer (BD) based Mach-Zehnder interferometer and generated the maximally entangled state $|\Phi^+\rangle = (|HH\rangle + e^{i\theta}|VV\rangle)/\sqrt{2}$. We set the relative phase $\theta$ through slight tilting of one of the BD. One of the photons from this source underwent a controllable depolarizing channel with probability $1 - \alpha$ that consisted of a Sagnac-based variable beam splitter (VBS) and a fully depolarizing operation[54]. A motorized HWP inside the interferometer controlled the splitting ratio of the VBS and was used to set the amount of mixture in the state. The fully depolarizing channel included two consecutive dephasing maps. Each of these used an imbalanced BD interferometer, generating a relative temporal delay between single-photon wavepackets with orthogonal polarization modes, correlating the polarization and time degree of freedom. An additional HWP set to 22.5° was inserted between the BD interferometers to dephase in two different bases. The photon detection does not resolve different arrival times, effectively resulting in a polarization mixture. The experimental states generated in this way were very similar to two-qubit isotropic states $W_\alpha$, although we do not make this assumption to reach our experimental conclusions. For various proportions of mixture, we reconstructed the density matrices of the experimentally produced state via maximum-likelihood quantum state tomography. A second photon source prepared a heralded photon to be used as an ancilla for the broadcast channel. Its polarization state was fixed.

### Experimental error analysis

We derived experimental uncertainties in the state parameter $\alpha$ and the locality certificate $\eta$ from tomographic reconstructions, considering systematic measurement errors and statistical errors intrinsic to probabilistic photon sources. Quantum state tomography, as is usual in device-dependent tasks, requires that the measurement devices used are precisely characterized and calibrated. It is thus crucial to avoid mischaracterizing our generated experimental states for our locality test algorithm. The systematic errors included: (i) imperfect calibration of the measurement wave plates, (ii) mechanical repeatability of the motorized stages involved in the measurement, and (iii) phase-shift errors due to manufacturing imperfections in wave plate thickness. We reconstructed 2000 density matrices for each experimental state in a Monte Carlo simulation, with each trial independently sampling the systematic (statistical) errors from a normal (Poissonian) distribution. For each reconstructed matrix, we calculated the parameters $\alpha$ and $\eta$, and the standard deviations of the distributions in the parameters produced the final uncertainties. Conversely, the broadcast inequality is a device-independent task and does not rely on the actual states used or measurements performed. The uncertainties in the inequality values were calculated from Poissonian photon counting statistics and standard error propagation techniques.

## Broadcast channel

After applying an appropriate broadcast channel on half of the original state, it is possible to obtain a quantum violation of inequality (3) for $W_\alpha$. This operation aims to map the two-dimensional Hilbert space of the original subsystem to another space of dimension $2^m$, where $m = 2$ is the total number of broadcast parties. In our case, the transformation carried out by the channel $\Omega_{BC}$ can be modeled as an isometry $V : \mathbb{C}^2 :\to \mathbb{C}^4$, decomposable into single-qubit rotations and C-NOT gates[55].

Our instance of a suitable broadcast channel was inspired by Ref. 24 and featured one probabilistic C-NOT gate[37] and single-qubit operations. The quantum circuit for this channel is shown in Fig. 2b, and the corresponding isometry is described by

$$V = \left(\frac{|HH\rangle - |VV\rangle}{\sqrt{2}}\right)\langle H| - \left(\frac{|HV\rangle + |VH\rangle}{\sqrt{2}}\right)\langle V|. \tag{18}$$

The inclusion of an additional ancillary photon was to physically enforce the no-signaling required between the broadcast parties, in contrast to using different degrees of freedom of a single photon[56].

Keeping the number of nondeterministic gates to a minimum is necessary since it can be impossible to verify the success of cascaded operations in post-selection. Since the ancilla photon state is fixed, we can use known methods to construct efficient isometries[57]. The single-qubit rotations used a combination of quarter-wave (QWP) and half-wave plates, and the C-NOT gate was implemented with partially polarizing beam splitters (PPBS). In our case, the C-NOT gate only required two PPBSs because of the fixed polarization state in the ancilla, raising the success probability from 1/9 to 1/6. To maximize the HOM interference visibility at the central PPBS, we placed bandpass filters (3.2 nm full width at half maximum) at each output port of the gate. We measured interference visibility of $0.97 \pm 0.03$ between photons generated from independent sources with no background subtraction.

## Projective measurements

A binary variable determined the choice of measurements for Bob and Charlie ($y, z = 0, 1$), while Alice had a choice of three possible measurements ($x = 0, 1, 2$). The settings that result in a maximal violation of inequality (3) with our broadcast channel are summarized in Table 1. An additional, fixed HWP was inserted before Alice's measurement station, rotating her measurements to {$\sigma_Z, \sigma_X, \sigma_Y$}, for experimental convenience.

Each party used a polarization measurement stage—consisting of a QWP, HWP, and PBS—to perform arbitrary projective measurements on their qubit. Photon events were detected with SNSPDs at both outputs of the PBS. Using a coincidence window of 1 ns, we collected 30633 four-fold coincidence events among all six data points in approximately 51 hours. For the data point corresponding to certified activation (Fig. 3, red triangle), we measured 9802 four-fold coincidences.

## Data availability

All data generated and analyzed during this study is available from the corresponding author upon reasonable request.

## Code availability

All the code used in this work is openly available at the following link: https://github.com/mtcq/LHVextention.

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

## Acknowledgements

This work was supported in part by ARC Grant No. DP210101651 (N.T., S.S., E.G.C. and G.J.P.) and in part by ARC Grant No. CE170100012 (S.R. and G.J.P.). The authors thank Matthias Kleinmann and H. Chau Nguyen for valuable discussions. L.V.-A. acknowledges support by the Australian Government Research Training Program (RTP).

## Author contributions

L.V.-A., S.S., N.T. and G.J.P. conceived the experiment. K.L. and E.G.C. adapted the theory to the experiment. L.V.-A. built the setup and carried out the experiment with help from S.S., F.G., E.P. and N.T. L.V.-A. and E.P. performed the data analysis and modeling with help from L.K.S. M.T.Q. derived the locality certification algorithm. I.R.B. and S.R. developed the high-efficiency SNSPDs. L.V.-A. wrote the manuscript with contributions from all authors. N.T., S.S. and G.J.P. supervised the experimental work, and E.G.C. supervised the theory work. All authors contributed to the discussion and interpretation of results.

## Competing interests

The authors declare no competing interests.
