## [Peer Review File · Nature Communications]

Nonlocality activation in a photonic quantum networkREVIEWER COMMENTS

Reviewer #1 (Remarks to the Author):

Bell scenario setup tool to distinguish LVH model from quantum mechanics through a set of correlations between inputs and outputs data collected distant parties. Not all the noisy entangled states will violate inequality and exhibit the nonlocality. The increase of the entanglement is possible with multicopy and LOCC or probabilistic with one copy. Similar schemes have been developed for the nonlocality activation.

Reference [24] have proposed the broadcast channel method. The authors demonstrate an experimental activation of nonlocality in a photonic quantum network using a single copy of the target state.

The theoretical and the experimental methods are well described, and clear written. The data analysis is good. The authors have used modern methods "semidefinite programming relaxations for quantum correlations". The results and conclusion are correct.

Before to recommend this paper, I would like to ask the authors to comment on:

1. how their imperfect HOM inference effect on their results.
2. how to reach loophole free version of the actual experiment
3. the authors used an extra photon as ancilla. Would you comment on instead of using an extra photon, one can (i) convert Bob photon to two photons through Nonlinear process? or (ii) two degrees of freedom of Bob's photon. (polarization and path).
4. one have to assume that the dimension of the prepared state Alice ($d=2$) and Bob ($d=2$).

Reviewer #2 (Remarks to the Author):

This work presents an experimental demonstration of the phenomenon of activation of nonlocality. Namely, the authors experimentally create a two-qubit state that they certify not to be able to violate any Bell inequality for dichotomic measurements, and (this is the important part of the work) a broadcasting channel that allows to reveal that nonlocality was actually present in the state. This is achieved by (1) certifying the inability of the two-

qubit state to violate any bipartite Bell inequality for dichotomic measurements using semidefinite optimization tools, and (2) using a Bell-like inequality, developed in Ref. [24] in the manuscript, that considers local models in the corresponding broadcasting scenario. Out of all the states prepared, for one of them the authors are able to certify activation.

I think that the manuscript is well written and the topic explored is very relevant at this moment. Multipartite scenarios are receiving much attention, and this work transcends the standard notion of quantum networks while showing advantages in a task that is known to require more complicated setups if done in the traditional quantum network paradigm. As the authors correctly identify, it takes a star network with 21 parties to demonstrate similar results (this number has been very recently reduced to 10, see arXiv:2310.20677, but it is still far of the method used by the authors). This work is a nice experimental demonstration, that will motivate more research in the area. The paper is clearly written, the experimental procedure seems sound, the error analysis is rigorous, and the conclusions are justified. For all this, I recommend publication of the manuscript in Nature Communications.

This being said, I have two comments for the authors to consider.

The first one is on a specific part in the writing. When discussing the preparation of states on lines 206-209 ("We prepared six experimental states ρ_{exp} and their average measured fidelity [...]"), the authors present as a measure of the quality of their preparations the average fidelity with ideal states, where the average is over the preparation of the (six) different states they consider in the work. I think that presenting the average fidelity over different states is somehow misleading, since they are treating on equal footing fidelities with different target states. Since the authors have the individual fidelities in the Supplementary Information, it suffices to direct the reader there, and perhaps to mention that all of them are above 0.991, which is still a very good value.

The second one concerns the data analysis. The "classical" model (Equation 2) that the authors consider has two elements, the LHV and the non-signaling distribution between Bob and Charlie. Therefore, the Bell-like inequality that they use to witness incompatibility with such a model (Equation 3) can be violated in two different ways: by having a non-local

quantum state in the source (as the authors do), or by having a procedure that creates a signaling distribution between Bob and Charlie. I acknowledge that enforcing no-signaling in this experiment is way beyond the scope of the work as it would require placing Bob and Charlie in spacelike-separated locations, but it would be useful to have an estimation of the degree up to which non-signaling is satisfied (or not) in the empirical data, for instance by showing the mutual information of the measurement outcomes of Bob and Charlie.

Finally, I want to note a typographic mistake in the Supplementary Information. On line 25, the closing parenthesis in the expression of t_{ij} is missing.

Reviewer #3 (Remarks to the Author):

In their manuscript, the authors report an experimental implementation of recent proposals of nonlocality activation based on a broadcasting map (fig 1b).

If no mistake (due to lack of time, I did not come back to these references), the theoretical works of ref [24,28] shown that it was possible to “activate nonlocality with broadcasting”.

More precisely, it is possible to:

- Start from a noisy maximally entangled state W_α (of eq (4)) which cannot create nonlocal correlations when measured with projective measurements (which is the case for $\alpha < 0.6875$)

- Broadcast it as in Fig 2.b and obtain a tripartite correlation (where A, B, C perform a projective measurement) which cannot be simulated in any “classical strategy”.

Here classical strategies are defined based on the model of eq.2, which models the broadcasting as the distribution of a NS box between B and C, that is in the most conservative model of broadcasting compatible with the no signaling principle.

In their experiment, the authors performed the corresponding experimental setup:

- a. they create (approximatively) some states W_α (for both $\alpha < 0.6875$, $\alpha > 0.6875$)
- b. they perform the tomography of these states, estimating ρ_{exp} . Then, they look for an LHV model for a possible dichotomic measurements of ρ_{exp} by looking for the existence of a decomposition of ρ_{exp} as a mixture of $W_{\alpha=0.6875}$ and a separable state (eq 10).
- c. they look for a violation of their broadcasting inequality (3)

For $\alpha \sim 0.64$, they obtain a ρ_{exp} which has an LHV model according to b., and violates the inequality as in c. They conclude activation of nonlocality for that state.

I find this manuscript well written and interesting. I however have the following questions and suggestions:

1. My first important remark is about the definition of the term “activation” and “local states”. The authors adopt the definition “ W_{α} is local if its correlations are local for any projective measurements”. Yet it is known that by considering more complex POVM measurements (with more inputs in the Bell scenario), it is possible to prove the nonlocality of more states. Hence, $W_{\alpha=0.6875}$ is (if no mistake) not proven to be local when more general POVM measurements (with more inputs) are considered. I am not an expert of this precise question, but I imagine that it is not known whether the state corresponding to $\alpha \sim 0.64$ admit a local model for all POVM measurements (with infinitely many inputs)? I think this should be more explicit in the manuscript.

2. I understand that adopting this definition of locality would result in a concept of “activation” completely out of reach theoretically/experimentally. In my opinion, it still makes sense to consider a more restricted question of finding “activation” in a more restrictive definition of “local states”, where less measurements are allowed to reveal nonlocality. However, one should in that case very carefully make sure the comparison between the “Bell scenario with projective measurements” and the “new scenario with activation” is fair. I’m not completely convinced it is the case here:

- in the Bell scenario in which the nonlocality is assessed, only qubit projective measurements are considered

- in the broadcast scenario, the experimental setup is based on new, different resources, in particular this broadcast channel in Fig 2b. Held by a single party, this channel could already be used to effectively perform POVM measurements. E.g., one could imagine that in Fig 1b Ω , B and C are together in a same location and perform a measurement using the ancilla $|H\rangle$ and the CNOT. This would allow them to effectively perform a POVM measurement of their share of W_{α} in a standard Bell scenario. And maybe see some nonlocality for $\alpha \sim 0.64$? I am interested in the author’s opinion on this point, which I find important: if the question is relevant and this was true, it would indicate that the “activation of nonlocality” seen by the authors is only coming from the fact that they go from PVM to

(some more general) POVMs and does not really use the broadcasting? I don't know how the conceptually justify that this is not what is happening.

2. In their discussion about hybrid scenarios before the method section, the authors could discuss [arXiv:2211.14231](https://arxiv.org/abs/2211.14231) and [arXiv:2310.07484](https://arxiv.org/abs/2310.07484)

I will be happy to read the author's reply and review their revised manuscript,

Marc-Olivier Renou

January 22, 2024

We thank the Reviewers for their careful analysis of our work, entitled *Nonlocality activation in a photonic quantum network* (manuscript number NCOMMS-23-46993-T), and for their valuable feedback. We provide detailed point-by-point replies to all the raised comments and suggestions. The Reviewers' remarks are in **blue font**, our replies in normal black font, and text modifications to the revised manuscript in **green font**.

Reviewer #1

Bell scenario setup tool to distinguish LVH model from quantum mechanics through a set of correlations between inputs and outputs data collected distant parties. Not all the noisy entangled states will violate inequality and exhibit the nonlocality. The increase of the entanglement is possible with multicopy and LOCC or probabilistic with one copy. Similar schemes have been developed for the nonlocality activation. Reference [24] have proposed the broadcast channel method. The authors demonstrate an experimental activation of nonlocality in a photonic quantum network using a single copy of the target state. The theoretical and the experimental methods are well described, and clear written. The data analysis is good. The authors have used modern methods "semidefinite programming relaxations for quantum correlations". The results and conclusion are correct.

We thank the Reviewer for their positive assessments of our work. The Reviewer poses some interesting technical questions, which we answer below.

Before to recommend this paper, I would like to ask the authors to comment on: 1. how their imperfect HOM inference effect on their results

We thank the Reviewer for pointing out that the effect of imperfect HOM interference might not be clear from the previous text. Having non-ideal HOM interference will introduce additional, unwanted mixture to the three-photon target state, which will reduce the observed value of the inequality (3). In Figure 3.a, the dashed gray diagonal line indicates the theory predictions considering ideal HOM interference, while the solid blue diagonal line takes into account the effects of imperfect interference, using our best estimate (0.97) of the experimental visibility. To make this more explicit, we have modified the manuscript as follows:

The observed experimental values are well captured by the predictions (Fig. 3a, solid diagonal line) derived from a theoretical model that considers non-ideal HOM interference (introducing unwanted mixed terms to the final target state) and errors in the performed measurements. We also include predictions for the case of ideal interference (Fig. 3a, dashed diagonal line).

2. how to reach loophole-version of the actual experiment

The main challenge to overcome would be closing the detection loophole. Our experimental demonstration features a broadcast channel that relies on a nondeterministic two-qubit gate with $1/6$ success probability. One way to increase the efficiency would be by using heralded gate schemes (Phys. Rev. Lett. 126, 140501 (2021)) or considering hybrid light-matter systems, such as photon-electron entanglement in previous loophole-free Bell tests (Nature 526, 682–686 (2015)).

Regarding the locality loophole, in our experiment, we already have all three parties spatially separated from one another. To close the locality loophole, it would be sufficient to increase these distances to space-like separations and use adequately fast measurements. Such measurements are possible for polarization-encoded photons via Pockels cells (see Phys. Rev. Lett. 115, 250402 (2015)).

Finally, to avoid the freedom-of-choice loophole, all three parties should make random measurements that are not influenced by each other or by the photons used in the experiment. A loophole-free version of our experiment would place all parties in space-like separation from the photon-pair sources too, where they may each independently choose their inputs based on randomized, on-demand fast signals that are close to their measurement stations. We have included the following perspective on this topic in the Discussion:

To achieve a fully loophole-free implementation, the key assumption to eliminate is the fair-sampling assumption. This would require increasing the overall efficiency, which is currently limited by the probabilistic implementation of the channel.

3. the authors used an extra photon as ancilla. Would you comment on instead of using an extra photon, one can (i) convert Bob photon to two photons through Nonlinear process?

We thank the Reviewer for bringing this interesting point to our attention. Indeed, our broadcast channel could be potentially implemented via a cascaded down-conversion process, for example, in nonlinear waveguides (Nature 466, 601–603 (2010)), but without closing the detection loophole.

or (ii) two degrees of freedom of Bob's photon. (polarization and path).

In principle, the broadcast channel could also be implemented using a different degree of freedom of Bob's photon as an ancillary qubit. However, in this case, it would be impossible to justify the no-signalling condition required between Bob and Charlie, given that they would be in the same physical location. This no-signalling condition is necessary to guarantee that the observed nonlocality does not originate from the broadcasting device alone. Our experiment considers this prerequisite by using different photons and spatially separated parties. In response to the Reviewer's comment, we have included additional information in the Methods:

Our instance of a suitable broadcast channel was inspired by Ref. [24] and featured one probabilistic C-NOT gate [35] and single-qubit operations. The inclusion of an additional ancillary photon was to physically enforce the no-signalling required between the broadcast parties, in contrast to using different degrees of freedom of a single photon [Phys. Rev. Lett. 95, 260501 (2005)].

4. one have to assume that the dimension of the prepared state Alice ($d=2$) and Bob ($d=2$).

We thank the Reviewer for the opportunity to clarify this point. Just like with Bell nonlocality, the certification of broadcast nonlocality makes no assumption about the Hilbert space dimensions of the systems or the characterisation of the devices; indeed not even the postulates of quantum theory are assumed to hold. In this sense, it is a theory-independent certification.

To clarify, in order to design the experiment, we apply our knowledge of quantum theory and optics, of course. However, for the derivation of the broadcast inequality, no assumptions are made about the dimension of the system or the specifics of any device involved in the experiment. The only assumptions involved at this point are (i) the existence of a Local Hidden Variable (LHV) model in the bipartition given by A and BC and (ii) that the bipartition given by B and C respects non-signalling constraints.

It is important to note that the only moment where assumptions about the devices are made is when performing quantum state tomography, where we assume our measurements are known and characterized. This naturally also includes the dimension of the system. We stress, however, that this dimensionality assumption is solely used to ensure that our reconstructed quantum state cannot violate a Bell inequality in the standard scenario, not to certify broadcast nonlocality in the experiment.

Reviewer #2

This work presents an experimental demonstration of the phenomenon of activation of nonlocality. Namely, the authors experimentally create a two-qubit state that they certify not to be able to violate any Bell inequality for dichotomic measurements, and (this is the important part of the work) a broadcasting channel that allows to reveal that nonlocality was actually present in the state. This is achieved by (1) certifying the inability of the two-qubit state to violate any bipartite Bell inequality for dichotomic measurements using semidefinite optimization tools, and (2) using a Bell-like inequality, developed in Ref. [24] in the manuscript, that considers local models in the corresponding broadcasting scenario. Out of all the states prepared, for one of them the authors are able to certify activation.

I think that the manuscript is well written and the topic explored is very relevant at this moment. Multipartite scenarios are receiving much attention, and this work transcends the standard notion of quantum networks while showing advantages in a task that is known to require more complicated setups if done in the traditional quantum network paradigm. As the authors correctly identify, it takes a star network with 21 parties to demonstrate similar results (this number has been very recently reduced to 10, see arXiv:2310.20677, but it is still far of the method used by the authors). This work is a nice experimental demonstration, that will motivate more research in the area. The paper is clearly written, the experimental procedure seems sound, the error analysis is rigorous, and the conclusions are justified. For all this, I recommend publication of the manuscript in *Nature Communications*.

We thank the Reviewer for their detailed assessment of our work and we are happy to see that they recommend its publication in *Nature Communications*. Additionally, we value the Reviewer's comments regarding the relevance of our results and the quality of our text and analysis. We thank the Reviewer for their feedback, which has improved the manuscript. In the new version of the manuscript, we are adding the reference arXiv:2310.20677 as Ref. [40].

This being said, I have two comments for the authors to consider. The first one is on a specific part in the writing. When discussing the preparation of states on lines 206-209 ("We prepared six experimental states ρ_{exp} and their average measured fidelity [...]"), the authors present as a measure of the quality of their preparations the average fidelity with ideal states, where the average is over the preparation of the (six) different states they consider in the work. I think that presenting the average fidelity over different states is somehow misleading, since they are treating on equal footing fidelities with different target states. Since the authors have the individual fidelities in the

Supplementary Information, it suffices to direct the reader there, and perhaps to mention that all of them are above 0.991, which is still a very good value.

We thank the Reviewer for pointing this out. To address this, we have modified the sentence in the following way:

We prepared six experimental states ρ_{exp} and their measured fidelities, defined as $\mathcal{F} = \text{Tr}(\sqrt{\sqrt{\rho_{\text{exp}}}W_\alpha\sqrt{\rho_{\text{exp}}}})^2$, with the nearest W_α state were all $\mathcal{F} > 0.991$ (see Supplementary Table I).

The second one concerns the data analysis. The ‘‘classical’’ model (Equation 2) that the authors consider has two elements, the LHV and the non-signaling distribution between Bob and Charlie. Therefore, the Bell-like inequality that they use to witness incompatibility with such a model (Equation 3) can be violated in two different ways: by having a non-local quantum state in the source (as the authors do), or by having a procedure that creates a signaling distribution between Bob and Charlie. I acknowledge that enforcing no-signaling in this experiment is way beyond the scope of the work as it would require placing Bob and Charlie in spacelike-separated locations, but it would be useful to have an estimation of the degree up to which non-signaling is satisfied (or not) in the empirical data, for instance by showing the mutual information of the measurement outcomes of Bob and Charlie.

We thank the Reviewer for raising this important observation. Experimental probabilities are derived from a finite number of samples, introducing unavoidable statistical fluctuations. Consequently, any non-signalling constraint can only be approximately satisfied, even in the case of space-like separation between parties. Following the Reviewer’s recommendation, we explicitly verified that our results are consistent with the no-signalling condition formalized by equations (7) and (8) in the Methods. To check this, we computed the expressions:

$$E_B^{\text{NS}}(y, y', z, c) = \left| \sum_b p_{\text{BC}}(b, c|y, z) - p_{\text{BC}}(b, c|y', z) \right| \quad \forall y, y', z, c, \quad (\text{R1})$$

$$E_C^{\text{NS}}(y, z, z', b) = \left| \sum_c p_{\text{BC}}(b, c|y, z) - p_{\text{BC}}(b, c|y, z') \right| \quad \forall y, z, z', b. \quad (\text{R2})$$

The no-signalling condition requires that $E_B^{\text{NS}} = E_C^{\text{NS}} = 0$. We computed the expressions (R1) and (R2) for all non-trivial combinations ($y \neq y', z \neq z'$). As the Reviewer can appreciate from Table R1, we found the mean value of these expressions to be $\langle E^{\text{NS}} \rangle = 0.02 \pm 0.05$, indicating that our results are consistent with no-signalling being satisfied. We have included these results as Supplementary Note III.

z	$E_B^{\text{NS}}(y \neq y')$	y	$E_C^{\text{NS}}(z \neq z')$
0	$1.66\text{e-}02 \pm 4.37\text{e-}02$	0	$3.75\text{e-}02 \pm 4.17\text{e-}02$
1	$2.96\text{e-}04 \pm 4.32\text{e-}02$	1	$1.63\text{e-}02 \pm 4.74\text{e-}02$

TABLE R1: No-signalling condition between Bob and Charlie

Finally, I want to note a typographic mistake in the Supplementary Information. On line 25, the closing parenthesis in the expression of t_{ij} is missing.

We thank the Reviewer for pointing out the typo. We have corrected this in the reviewed manuscript.

Reviewer #3

In their manuscript, the authors report an experimental implementation of recent proposals of nonlocality activation based on a broadcasting map (fig 1b). If no mistake (due to lack of time, I did not come back to these references), the theoretical works of ref [24,28] shown that it was possible to ‘‘activate nonlocality with broadcasting’’. More precisely, it is possible to:

- Start from a noisy maximally entangled state W_α (of eq (4)) which cannot create nonlocal correlations when measured with projective measurements (which is the case for $\alpha < 0.6875$)
- Broadcast it as in Fig 2.b and obtain a tripartite correlation (where A, B, C perform a projective measurement) which cannot be simulated in any ‘‘classical strategy’’.

Here classical strategies are defined based on the model of eq.2, which models the broadcasting as the distribution of a NS box between B and C, that is in the most conservative model of broadcasting compatible with the no signaling principle. In their experiment, the authors performed the corresponding experimental setup:

a. they create (approximatively) some states W_α (for both $\alpha < 0.6875$, $\alpha > 0.6875$)
 b. they perform the tomography of these states, estimating ρ_{exp} . Then, they look for an LHV model for a possible dichotomic measurements of ρ_{exp} by looking for the existence of a decomposition of ρ_{exp} as a mixture of $W_\alpha = 0.6875$ and a separable state (eq 10).

c. they look for a violation of their broadcasting inequality (3)

For $\alpha \sim 0.64$, they obtain a ρ_{exp} which has an LHV model according to b., and violates the inequality as in c. They conclude activation of nonlocality for that state. I find this manuscript well written and interesting.

We thank the Reviewer for their comprehensive assessment of our manuscript and for their interest in our work.

Below, we provide a point-by-point response to the detailed comments of the Reviewer, which we used to improve the quality of our manuscript. We hope that the Reviewer will be satisfied with our changes and recommend publication in *Nature Communications*.

I however have the following questions and suggestions:

1. My first important remark is about the definition of the term “activation” and “local states”. The authors adopt the definition “ W_α is local if its correlations are local for any projective measurements”. Yet it is known that by considering more complex POVM measurements (with more inputs in the Bell scenario), it is possible to prove the nonlocality of more states. Hence, $W_\alpha = 0.6875$ is (if no mistake) not proven to be local when more general POVM measurements (with more inputs) are considered. I am not an expert of this precise question, but I imagine that it is not known whether the state corresponding to $\alpha \sim 0.64$ admit a local model for all POVM measurements (with infinitely many inputs)? I think this should be more explicit in the manuscript.

We appreciate the Reviewer’s insightful comments and agree that further clarification is necessary. We have revised our manuscript to provide a more detailed discussion of these aspects.

As correctly noted by the Reviewer, the two-qubit Werner state W_α is known to have an LHV model for projective measurements (and indeed for arbitrary dichotomic POVMs) for $\alpha \leq 0.6875$. When more general POVMs are considered, very recent works (Refs. [arXiv:2309.09960, arXiv:2309.12290]) establish that W_α is Bell local for $\alpha \leq 0.5$. It remains an important open question in the field whether this state is nonlocal under more general POVMs in the range $0.5 < \alpha \leq 0.6875$. We have included this information in the revised manuscript to make it clear and explicit:

Here, the parameter $\alpha \in [0, 1]$ is the pure-state fraction of the state, which cannot display Bell nonlocality for $\alpha < 0.6875$ under dichotomic measurements (see [33] and Methods). For general measurements, the current known bound is $\alpha \leq 0.5$ [arXiv:2309.09960v1, arXiv:2309.12290v1].

However, in the case of two-qubit states, and particularly for two-qubit Werner states, there is evidence suggesting that general POVMs may not be useful for revealing Bell nonlocality in the standard Bell scenario.

Before discuss this point below, we want to make a comment on a particular statement by the Reviewer:

“Yet it is known that by considering more complex POVM measurements (with more inputs in the Bell scenario), it is possible to prove the nonlocality of more states.”

This is a subtle issue, and whether this statement is correct depends on how it is interpreted. For some specially tailored Bell inequalities, POVMs may allow for its violation by some states that do not violate that inequality with projective measurements, as can be shown from the results of Ref. [Phys. Rev. A 82, 062115 (2010)]. In this narrow sense, POVMs prove the nonlocality of more states. We note, however, that the relevant question in our case is whether, for a state that is known to possess an LHV model under projective measurements, there exists a Bell inequality that it can violate when considering more general POVMs. This is a crucial open question in quantum information science.

We believe that the prevailing view in the field is that POVMs provide no advantage over projective measurements to reveal the nonlocality of quantum states in this relevant sense. Specifically, if there exists an LHV model for projective measurements, it seems there is an LHV model for general POVMs as well. This observation has very recently been proven to hold true for two-qubit states W_α in the case of EPR steering, for the class of Local Hidden State (LHS) models, where one party is assumed to perform quantum response functions (see Refs. [arXiv:2309.09960, arXiv:2309.12290] mentioned above for a proof). For this kind of nonlocality, POVMs and projective measurements are equivalent. Moreover, there is evidence (Ref. [Phys. Rev. A 96, 022110 (2017)]) indicating that the most incompatible sets of qubit POVMs are always projective.

2. I understand that adopting this definition of locality would result in a concept of “activation” completely out of reach theoretically/experimentally. In my opinion, it still makes sense to consider a more restricted question of finding “activation” in a more restrictive definition of “local states”, where less measurements are allowed to reveal nonlocality. However, one should in that case very carefully make sure the comparison between the “Bell scenario with projective measurements” and the “new scenario with activation” is fair. I’m not completely convinced it is the case here: - in the Bell scenario in which the nonlocality is assessed, only qubit projective measurements are considered - in the broadcast scenario, the experimental setup is based on new, different resources, in particular, this broadcast channel in Fig 2b. Hold by a single party, this channel could already be used to effectively perform

POVM measurements. E.g., one could imagine that in Fig 1b Ω , B and C are together in the same location and perform a measurement using the ancilla $|H\rangle$ and the CNOT. This would allow them to effectively perform a POVM measurement of their share of W_α in a standard Bell scenario. And maybe see some nonlocality for $\alpha \sim 0.64$? I am interested in the author’s opinion on this point, which I find important: if the question is relevant and this was true, it would indicate that the “activation of nonlocality” seen by the authors is only coming from the fact that they go from PVM to (some more general) POVMs and does not really use the broadcasting? I don’t know how they conceptually justify that this is not what is happening.

We thank the Reviewer for raising this point. The Reviewer accurately points out that, in the broadcast scenario, the channel and the measurements performed by parties B and C can be seen as an effective POVM. Formally, one would consider the behavior:

$$p(a, b, c|x, y, z) = \text{Tr}(\rho_{ABC} A_{a|x} \otimes M_{bc|yz}) = \text{Tr}(\rho_{AB} A_{a|x} \otimes \Omega^\dagger(M_{bc|yz})), \quad (\text{R3})$$

where Ω is the broadcast channel mapping one qubit into two qubits, and $\Omega^\dagger(M_{bc|yz})$ is a valid qubit 4-outcome POVM performed by B and C grouped together after the channel.

Firstly, we remark it has been explicitly shown in Ref. [SciPost Phys. Core 6, 028 (2023)], a previous work by one of the authors, that the broadcasting scenario can indeed allow for activation with states that are known to be Bell-local for general POVMs. This establishes that the advantage provided by the broadcasting scenario cannot be understood in general as simply arising from the effective use of POVMs.

Furthermore, we can numerically verify that the measurements we actually performed in the experiment cannot result in a Bell nonlocal behavior in this way. In a new section that we have added to the Methods, we show that for the particular state we prepared and measurements performed at B and C, it is not possible to demonstrate EPR steering, and as a consequence, it is not possible to demonstrate Bell nonlocality across the A/BC partition – even allowing arbitrary POVMs at Alice’s side.

To prove this, we use standard semidefinite programming methods for EPR steering and joint measurability (Ref. [Rep. Prog. Phys. 80, 024001 (2017)]) to show that our “effective POVMs” $\Omega^\dagger(M_{bc|yz})$ have a white noise robustness of 0.7746. In other words, when such measurements are performed by parties BC on their part of W_α for $\alpha \leq 0.7746$, the resulting assemblage is unsteerable. To confirm that this is the case for our experimental state ρ_{exp} , we reconstructed the resulting assemblage

$$\sigma_{bc|yz} := \text{Tr}_B(\rho_{\text{exp}} \Omega^\dagger(M_{bc|yz})). \quad (\text{R4})$$

We verified that this assemblage is also unsteerable, which implies that it can only lead to Bell local behaviors, regardless of what measurements Alice performs. This new analysis establishes unambiguously that the observed nonlocality in our experiment is not a consequence of transitioning from projective measurements to POVMs, which we hope addresses the specific concern raised by the Reviewer. The code for this calculation is publicly available at the following link: <https://github.com/mtcq/LHVextension/blob/main/EnsureEffectivePOVMisLocal.m>. We sincerely thank the Reviewer for raising this point, as it led us to new analysis that we used to improve the presentation of our results in the manuscript.

Finally, we agree with the Reviewer that requiring proof of a LHV model for general POVMs would result in a concept of “activation” completely out of reach theoretically/experimentally, and we’d like to take the opportunity to elaborate on this point. As discussed above, the causal constraints imposed by the broadcast scenario allow for a genuinely novel form of activation, which cannot be understood as simply arising out of effective 4-outcome POVMs being applied on the BC side. With this clarified, the comparison between the Bell and Broadcasting scenarios is also fair in that in both cases the local measurements are dichotomic. Importantly, our results achieve nonlocality activation from a practical viewpoint, in the sense that we demonstrated the nonlocality of states for which there is no known method of experimentally revealing their single-copy nonlocal nature in a standard Bell scenario.

3. In their discussion about hybrid scenarios before the method section, the authors could discuss [arXiv:2211.14231](https://arxiv.org/abs/2211.14231) and [arXiv:2310.07484](https://arxiv.org/abs/2310.07484)

We thank the Reviewer for their suggestions on relevant literature. In the new version of our manuscript, we mention these references as examples of scenarios with hybrid assumptions.

REVIEWERS' COMMENTS

Reviewer #1 (Remarks to the Author):

The authors have replied to my comments and added the relevant text

I recommend this work for publication for nature communications

Reviewer #2 (Remarks to the Author):

In this reviewed version of the manuscript, the authors satisfactorily address all my comments and concerns from previous versions. I thus recommend the article is published without further modifications.

Reviewer #3 (Remarks to the Author):

I am fully satisfied by the detailed answer of the authors and their modification of the manuscript.

I am happy to recommend publication in Nature Communications.